# Green Roof Substrate Microbes Compose a Core Community of Stress-Tolerant Taxa

**DOI:** 10.3390/microorganisms12071261

**Published:** 2024-06-21

**Authors:** Thomas Van Dijck, Vincent Stevens, Laure Steenaerts, Sofie Thijs, Carmen Van Mechelen, Tom Artois, François Rineau

**Affiliations:** 1Centre for Environmental Sciences, Zoology: Biodiversity and Toxicology, Hasselt University, 3590 Diepenbeek, Belgium; thomas.vandijck@uhasselt.be (T.V.D.); tom.artois@uhasselt.be (T.A.); 2Centre for Environmental Sciences, Environmental Biology, Hasselt University, 3590 Diepenbeek, Belgiumsofie.thijs@uhasselt.be (S.T.); 3PXL BIO-Research, 3590 Diepenbeek, Belgium

**Keywords:** bacteria, fungi, microbes, urban, green roof, aeromicrobiome

## Abstract

Extensive green roofs provide for many ecosystem services in urban environments. The efficacy of these services is influenced by the vegetation structure. Despite their key role in plant performance and productivity, but also their contribution to nitrogen fixation or carbon sequestration, green roof microbial communities have received little attention so far. No study included a spatiotemporal aspect to investigate the core microbiota residing in the substrates of extensive green roofs, although these key taxa are hypothesized to be amongst the most ecologically important taxa. Here, we identified the core microbiota residing in extensive green roof substrates and investigated whether microbial community composition is affected by the vegetation that is planted on extensive green roofs. Eleven green roofs from three different cities in Flanders (Belgium), planted either with a mixture of grasses, wildflowers and succulents (*Sedum* spp.; *Sedum*–herbs–grasses roofs) or solely species of *Sedum* (*Sedum*–moss roofs), were seasonally sampled to investigate prokaryotic and fungal communities via metabarcoding. Identifying the key microbial taxa revealed that most taxa are dominant phylotypes in soils worldwide. Many bacterial core taxa are capable of nitrogen fixation, and most fungal key taxa are stress-tolerant saprotrophs, endophytes, or both. Considering that soil microbes adapted to the local edaphic conditions have been found to improve plant fitness, further investigation of the core microbiome is warranted to determine the extent to which these stress-tolerant microbes are beneficial for the vegetational layer. Although *Sedum*–herbs–grasses roofs contained more plant species than *Sedum*–moss roofs, we observed no discriminant microbial communities between both roof types, likely due to sharing the same substrate textures and the vegetational layers that became more similar throughout time. Future studies are recommended to comprehensively characterize the vegetational layer and composition to examine the primary drivers of microbial community assembly processes.

## 1. Introduction

Green roofs, or vegetated roofs, are increasingly recognized as valuable urban green infrastructures due to their contribution to a number of ecosystem services in built-up environments. For example, green roofs retain stormwater runoff [1,2,3], remove air pollution [4], cool down building interiors [5,6], mitigate heat island effects [7,8], and conserve urban biodiversity [9,10,11]. The extent to which green roofs contribute to the above-mentioned ecosystem services depends on the characteristics of the substrate layer and the vegetation structure. 

Traditionally, green roofs are classified based on the thickness and composition of the substrate layer, the type of vegetation, and their primary function [12]. Most of the installed green roofs are “extensive” green roofs, which comprise thin layers (<15 cm) of porous mineral substrate low in organic matter (<10%). They are primarily installed to retain stormwater runoff, and their lightweight substrates offer the advantage that the roofs do not require costly structural reinforcements to the buildings. The composition of such substrates, however, results in a vegetational layer that needs to be adapted to the xerothermic conditions that periodically arise [13,14]. As a result, extensive green roofs are typically planted with either monocultures of succulents (species of *Sedum*) or succulents in combination with xerotolerant herbaceous species and grasses [15].

Green roof plant communities are not only shaped by the physio-chemical parameters of the substrate [16,17,18] but they will also be affected by the associated microbes in and around their roots. Soil microbial communities are known to vary in response to plant composition. Plant life forms, determined by plant history traits, play a significant role in shaping microbial community composition, which can be attributed to factors such as microbial-host specificity [19,20,21] or plant root structural differences [22]. Thus, it is expected that a greater plant diversity on extensive green roofs will result in more diverse and enriched microbial communities. Furthermore, soil bacteria and fungi are known to affect plant communities. For example, they shape plant diversity [23,24,25], contribute to overall plant performance and productivity [26,27,28], and influence plant functional traits [29,30]. Consequently, in green roofs, microbes can improve plant resilience to stress induced by frequent periods of drought and heat [31,32], resulting in improved ecosystem services. In addition to supporting the vegetational layer, microbes are integral to many other ecosystem services such as nitrogen-fixation [33,34], carbon sequestration [35,36], or decomposition [37], further underscoring the importance of examining microbial diversity in green roof substrates. 

While numerous studies have examined the application of inoculants in extensive green roof substrates to act as biostimulants [38,39,40,41,42,43,44], there has been comparatively limited research into identifying the trends that shape microbial communities within these novel urban habitats. The few studies that investigated green roof microbial assemblages via metabarcoding revealed that the communities are diverse and compositionally distinct from other urban green infrastructures, such as parks or bioswales [45,46,47,48]. Community assembly is believed to be primarily driven by environmental filtering rather than stochastic processes [46], although the surrounding environment has also been found to play an integral role in structuring green roof microbial communities [49]. Microbial communities have been linked to the roof vegetation in some studies [50] but not in all [45]. This is not surprising, considering the variation in green roof construction. More research is warranted to elucidate the main drivers behind microbial community assemblages in the substrates of green roofs.

Furthermore, no study to date addressed the temporal differences in the microbial community composition from green roof substrates, and most were geographically limited to investigating green roofs in one city (except [47,51]). Including a spatiotemporal aspect in microbiome studies allows for the identification of the core microbiota, i.e., a set of taxa that consistently occur within green roof substrates. Given that these shared taxa are hypothesized to be linked to host-plant functional traits [52], aid plants in nutrient acquisition and stress tolerance [53] and facilitate plant growth and performance [54,55], revealing the core microbiota and its ecological roles are crucial for understanding the ecosystem services delivered by green roofs in urban environments.

Here, we add to the limited body of knowledge about microbial communities in green roof substrates by employing 16S ribosomal RNA (Prokaryota) and internal transcribed spacer 2 (ITS2; Fungi) metabarcoding. Eleven extensive green roofs were selected across three cities in Flanders (Belgium) and sampled four times (three-monthly: once every season) to identify the core microbial taxa residing in the substrates. Furthermore, we selected green roofs that were either planted with species of *Sedum* or a combination of *Sedum*, wildflowers and grasses to examine whether green roofs having a higher plant richness contain discriminant microbial communities.

## 2. Materials and Methods

### 2.1. Study Area and Green Roof Characteristics

This study was performed in the Flemish Region of Belgium, Western Europe. The area has a temperate maritime climate. Throughout the duration of the study, a mean annual temperature of 11.5 °C was recorded, and the average rainfall amounted to 798.6 mm/m^2^ [56]. We selected eleven extensive green roofs (Table 1) in three cities within 100 km away from each other: Antwerp (51.2194° N, 4.4024° E), Ghent (51.0543° N, 3.7174° E) and Hasselt (50.9307° N, 5.3325° E). They are constructed on flat roofs using commercial extensive green roof substrate, i.e., mineral substrate (mainly pumice and crushed brick) low in organic matter content (<10%) (Appendix A). The roofs differ in the vegetation that was initially planted: either only species of *Sedum* (*Sedum*–moss roofs) or species of *Sedum* combined with wildflowers and grasses (*Sedum*–herbs–grasses roofs) [12]. All roofs differ in the size of their vegetated areas (25–777 m^2^), their year of construction (4–15 years before the start of our study), and the height at which they are located (3.2–22.3 m above ground level). However, the average values of these variables do not differ between both roof types.

### 2.2. Data Collection and DNA Extraction

Plant species richness on all green roofs was assessed by identifying the growing forbs and grasses to the species level in May, June, August and September 2019. Samples for microbial analyses were collected on three consecutive days (one day per city) on four occasions (once per season): in April, July and October 2019, and in January 2020. Three samples were collected from every roof in April, except for roof 11 (R11; not accessible at that moment). In the remaining months, four samples were taken per roof, resulting in a total of 162 samples.

The samples were collected by randomly placing plastic squares of 30 × 30 cm on every roof (Figure 1). Photographs were taken to assess the percentage cover of (i) *Sedum*, (ii) wildflowers and grasses, and (iii) bare substrate. Prior to sampling the substrate, the vegetational layer was removed. For each sample, the top 5 cm of substrate was collected and transferred to sterile plastic bags. Subsequently, the total substrate depth was measured within the plots. The substrate samples were kept on ice during transportation and stored at 4 °C for one week until further processing. 

Substrate samples were sieved in the lab at 2 mm. Quadruplicate aliquots of 250 mg were weighed and transferred to Eppendorf tubes, resulting in 648 samples. These samples were stored at −21 °C until DNA extraction, which was performed using DNeasy PowerSoil Pro kits (Qiagen, Hilden, Germany) following the manufacturer’s protocol. One empty disruption tube was included to serve as a negative control, while another tube containing the ZymoBIOMICS Microbial Community DNA Standard (Zymo Research, Irvine, CA, USA) was included as a positive control.

### 2.3. Metabarcoding Substrate Microbial Communities

Prokaryotic communities were investigated by amplifying the hypervariable V4 region of the 16S rRNA gene using the primers 515F (5′-GTGYCAGCMGCCGCGGTAA) and 806R (5′-GGACTACHVGGGTWTCTAAT) [57]. Fungal communities were examined via the hypervariable internal transcribed spacer 2 (ITS2) region between the 5.8S and 28S rRNA genes using the primers gITS86F (5′-GTGARTCATCGARTCTTTGAA) and ITS4R (5′-TCCTCCGCTTATTGATATGC) [58]. All primers had Nextera transposase adapters (Illumina, San Diego, CA, USA) attached to them. PCR reactions were conducted in 25 μL containing 1× Q5 reaction buffer (New England Biolabs, Ipswich, MA, USA), 200 μM dNTPs, 0.25 μM of both primers, 0.02 U/μL Q5 Hot-Start High Fidelity DNA polymerase, and 1 μL of DNA template. The thermocycling conditions consisted of an initial denaturation at 98 °C for 150 s—28 cycles of 10 s at 98 °C, 30 s at 55/57 °C (Prok/Fun), 30 s at 72 °C—a final extension for 7 min at 72 °C. Downstream detection of the PCR products was carried out by agarose gel electrophoresis, and quadruplicate samples were pooled based on the intensity of the bands. Pooled PCR products were purified using AMPure XP beads (Beckman Coulter, Brea, CA, USA) and subsequently indexed using Nextera XT indices (Illumina, San Diego, CA, USA). The 25 μL PCR mixtures contained 1× Q5 reaction buffer, 200 μM dNTPs, 0.02 U/μL Q5 Hot-Start High Fidelity DNA polymerase, 2.5 μL of both indices, and 1 μL of DNA template. Amplification conditions were as follows: 98 °C for 3 min—11× (98 °C for 10 s/55 °C for 30 s/72 °C for 30 s)—72 °C for 7 min. Indexed samples were purified (as described above) and quantified with a Qubit dsDNA HS assay kit on a Qubit 2.0 fluorometer (Thermo Fisher Scientific, Waltham, MA, USA). Purified indexed samples were brought to equimolar concentration prior to pooling. The final library was diluted to a concentration of 4 nM and sequenced 2 × 300 bp using a MiSeq Reagent Kit v3 (Illumina, San Diego, CA, USA) on a MiSeq system at Hasselt University (Belgium).

### 2.4. Data Analysis

R v4.1.2 [59] was used to perform the analyses. First, primers were removed from the raw amplicon sequencing data via the R-package cutadapt v2.9 [60]. Next, reads were processed into exact amplicon sequence variants (ASVs or OTUs generated using 100% sequence similarity) via dada2 v1.22.0 [61]. Based on the quality profiles of the reads, we truncated the forward reads at position 230/230 (Prokaryota/Fungi) and the reverse reads at position 120/160. The maximum number of expected errors was set to 1, and the number of ambiguous nucleotides to 0. We used standard parameters in all subsequent steps (error rate learning, sample inference, merging and chimera removal). Unique ASVs were aligned against the UNITE v9.0 reference database for Fungi [62] or SILVA v138.1 for Bacteria and Archaea [63]. The R-package phyloseq v1.38.0 [64] was used to merge the ASV-feature table, taxonomy table and table containing the metadata. 

Prior to performing the analyses, we pre-processed the data. Control samples (negative controls and mock communities) were investigated and discarded from the dataset. ASVs that were not assigned at the level of phylum were discarded since many were found to belong to plant species when blasted against GenBank [65]. Bacterial ASVs identified as chloroplasts or mitochondria were also discarded. Samples were rarefied to account for differences in library size following the investigation of sequencing depth via rarefaction curves through the R-package vegan v2.5.7 [66].

We compared the microbial communities of both roof types by examining alpha- and beta-diversity metrics. Regarding alpha-diversity, we calculated three commonly used metrics, i.e., ASV-richness (the number of ASVs per sample) and Shannon index (which also considers the proportions of the ASVs), which were calculated via vegan, and Faith’s phylogenetic diversity (PD; the sum of the branch lengths of the phylogenetic tree connecting all ASVs in the sample) via picante v.1.8.2 [67]. Significant differences in all three metrics between both roof types were measured via Wilcoxon rank-sum tests. The microbial community compositions (beta-diversity) of both roof types were compared via principal coordinate analysis (PCoA), performed on Weighted UniFrac distance matrices [68], which accounts for the phylogenetic relationship among the ASVs and uses ASV abundances to weigh the branch lengths. The effect of roof type on community composition was tested using permutational analysis of variance (PERMANOVA, R-package vegan) after validating the homogeneity of dispersion. Figures were made via ggplot2 v3.3.6 [69].

To further characterize the microbial communities, we identified key (core and discriminant) microbial taxa. Core taxa were considered to be ASVs that are present in at least 75% of all samples taken, having a relative abundance of at least 0.1%. Discriminant taxa were investigated via linear discriminant analysis (LDA) effect size (LEfSe) via the R-package microbiomeMarker v1.3.3 [70]. Prior to running the LEfSe analyses, we transformed the rarefied table to total sum scaling (relative abundance) and normalized the sum of the values to 1 M. We also added an individual identity to each ASV to prevent the script from merging all ASVs belonging to the same genus. Wilcoxon rank-sum tests (cutoff at 0.01) were used to identify differentially abundant features between both roof types, and linear discriminant analysis (LDA) estimated the effect size for the significant features (threshold score of 4.0).

## 3. Results

### 3.1. Sedum–moss Roofs vs. Sedum–herbs–grasses Roofs

We identified all growing plant species on the eleven extensive green roofs on four occasions in the growth season (May, June, August, and September 2019) (Appendix A). When summed up, *Sedum*–herbs–grasses roofs have significantly more herbs and grasses than *Sedum*–moss roofs (*p*-value 0.014; Figure 2a). The vegetational layers were further characterized by measuring the plant coverage (excluding mosses) on top of the plots that were sampled for the microbial analyses. On average, total plant coverage between both roof types does not differ (*p*-value 0.202; Figure 2b), but herbs and grasses account for a higher percentage on *Sedum*–herbs–grasses roofs (*p*-value < 0.001; Figure 2c). Simultaneously, we measured the substrate depth of the plots. The substrate layer is, overall, significantly thicker in *Sedum*–herbs–grasses roofs compared to *Sedum*–moss roofs (*p*-value < 0.001, Figure 2d).

### 3.2. Prokaryotic Diversity

Samples were rarefied to 4352 reads to account for uneven sampling depths (Appendix A). Eight samples were excluded from the dataset due to an insufficient number of reads. Across the remaining 154 samples (670,208 reads), 8308 prokaryotic ASVs (of which 78 archaeal) are discovered (average amplicon length 253 ± 2 bp). The taxonomic classification of the inferred ASVs via the SILVA database reveals three dominant phyla across all investigated green roofs, i.e., Proteobacteria (31.4%; 1738 ASVs), Actinobacteriota (23.8%; 1167 ASVs) and Acidobacteriota (14.5%; 774 ASVs) (Figure 3a). Extending the results to the ten most prevalent prokaryotic phyla revealed six more bacterial phyla [Bacteroidota (5.8%; 1119 ASVs), Chloroflexi (5.2%; 654 ASVs), Verrucomicrobiota (4.0%; 403 ASVs), Gemmatimonadota (4.0%; 398 ASVs), Myxococcota (3.8%; 377 ASVs), Planctomycetota (3.4%; 797 ASVs)] and one archaeal phylum: Crenarchaeota (1.3%; 47 ASVs).

None of the calculated alpha-diversity metrics differ statistically significantly between *Sedum*–moss roofs (n = 72) and *Sedum*–herbs–grasses roofs (n = 82), i.e., ASV richness (*p*-value 0.73), Shannon-index (*p*-value 0.99) and Faith’s phylogenetic diversity (*p*-value 0.30) (Figure 4a–c). The prokaryotic communities of both roof types differ compositionally significantly (*p*-value 0.001, R^2^ 0.062) and samples taken from *Sedum*–moss roofs have more variation in their community compositions than samples from *Sedum*–herbs–grasses roofs (*p*-value 0.003) (Figure 4d).

Regarding the core prokaryotic taxa, 16 taxa are found to be present in at least 75% of all samples, having a relative abundance ≥ 0.1%: 15 bacterial core taxa (Proteobacteria: *Massilia*, *Ramlibacter*, *Devosia*, *Rhodoplanes*, *Bradyrhizobium* and two ASVs within Xanthobacteraceae; Actinobacteriota: *Gaiella*, *Solirubrobacter*, *Streptomyces*, *Blastococcus*, *Pseudarthrobacter*, *Kineosporia*, and *Pseudonocardia*; Chloroflexi: one ASV within KD4-96) and 1 archaeal core taxon (Chrenarchaeota: *Candidatus Nitrocosmicus*) (Figure 5a). *Pseudarthrobacter* sp. is the only ASV found to be discriminant between both roof types, being more enriched in *Sedum*–herbs–grasses roofs.

### 3.3. Fungal Diversity

Samples were rarefied to 2331 reads (Appendix A) to account for uneven sampling depths. Eight samples were excluded from the dataset due to an insufficient number of reads. Across the remaining 154 samples (358,974 reads), 2671 fungal ASVs are discovered (average amplicon length 273 ± 33 bp). The taxonomic classification of the inferred ASVs via the UNITE database reveals one prevalent fungal phylum across all investigated green roofs, i.e., Ascomycota (84.0% of all reads; 1683 ASVs), followed by Basidiomycota (10.4%; 587 ASVs), Mucoromycota (2.0%; 11 ASVs), Chytridiomycota (1.6%; 188 ASVs), Mortierellomycota (1.4%; 54 ASVs) and Rozellomycota (0.4%; 44 ASVs) (Figure 3b).

Fungal diversity analyses follow similar patterns as those observed for the prokaryotic communities. None of the calculated alpha-diversity metrics differ statistically significantly between *Sedum*–moss roofs (n = 69) and *Sedum*–herbs–grasses roofs (n = 85), i.e., ASV richness (*p*-value 0.06), Shannon-index (*p*-value 0.37) and Faith’s phylogenetic diversity (*p*-value 0.80) (Figure 4e–g). The fungal communities of both roof types differ compositionally significantly (*p*-value 0.001, R^2^ 0.070), but in contrast to the prokaryotic communities, fungal communities of *Sedum*–moss roofs have more variation in their compositions than samples from *Sedum*–herbs–grasses roofs (*p*-value < 0.001) (Figure 4h).

Regarding the core fungal taxa (Figure 5b), four fungal ASVs are recovered in at least 75% of all samples, having a relative abundance ≥ 0.1%. These are all Ascomycota, i.e., *Arxiella*, *Cladosporium*, *Paraphoma*, and an ASV within the order Pleosporales. Nine taxa were found to be discriminant: ASVs within *Rhizopus*, *Coprinellus*, *Ascobolus* and *Fusarium* are more enriched in *Sedum*–herbs–grasses roofs, while ASVs within *Arxiella*, *Alternaria*, *Paraphoma*, *Stagonosporosis* and an ASV belonging to Pleosporales are more abundant in *Sedum*–moss roofs.

## 4. Discussion

Our study aimed to characterize the microbial communities residing in the substrates of extensive green roofs by investigating the presence of a core microbiota (ASVs). Furthermore, since species of *Sedum* do not root deeply into the substrate [71] and many plant species have host-specific microbes in their rhizobiomes [72,73], we investigated whether extensive green roofs planted with a combination of succulents (species of *Sedum*), flowers and grasses (i.e., *Sedum*–herbs–grasses roofs) harbor microbial communities that are more enriched and compositionally different from those in extensive green roofs solely planted with species of *Sedum* (i.e., *Sedum*–moss roofs).

Overall, prokaryotic communities in extensive green roof substrates are dominated by three bacterial phyla, i.e., Proteobacteria, Actinobacteriota, and Acidobacteriota. This corresponds to the results obtained in previous green roof studies [46,49]. Furthermore, the relative abundances of the ten major bacterial phyla in our study largely comply with their respective distributions in soils globally [74,75]. At the level of phylum, the fungal communities are dominated by Ascomycota, which also corresponds to previous green roof papers [45,46,50]. Furthermore, Ascomycota has been found to be the most phylotype-rich and abundant lineage in different biomes across the world [76]. Investigating the core microbiota revealed four fungal and 15 prokaryotic (one archaeal and 14 bacterial) core taxa. Although few bacterial taxa are typically shared between any pair of unique soil samples [77,78], the number of retrieved core microbial taxa in this study is relatively low. Currently, a wide range of metrics for quantifying the core microbiota are used. While methods that combine abundance and occurrence are well-supported in the ecological literature [79], primer selection, variation in sequencing depth and rarefying the datasets plausibly affected our results. Careful consideration of the approach to investigate core microbiota is advised for future studies [79].

Regarding the bacterial core community, a significant portion is composed of ASVs within the orders Rhizobiales (*Devosia*, *Rhodoplanes*, *Bradyrhizobium* and two ASVs within Xantobacteraceae) and Burkholderiales (*Massilia* and *Ramlibacter*), taxa that have also been found in a previous green roof study [49]. Furthermore, that study recovered abundant nitrogenase (nifH) genes affiliated with Rhizobiales, which suggests active nitrogen fixation by these strains. Except for the ASVs assigned to Chloroflexi, *Gaiella*, and *Pseudoarthrobacter*, all remaining bacterial core taxa identified in our study (*Solirubrobacter*, *Streptomyces*, *Blastococcus*, *Kineosporia* and *Pseudonocardia*) are dominant in soils globally [74]. Within the fungal communities, all identified core taxa in our study are Ascomycota, i.e., *Cladosporium*, *Arxiella*, *Paraphoma*, and an ASV within the order Pleosporales. The dominance of species within Ascomycota is plausibly attributed to a higher number of genes associated with stress tolerance and resource uptake, indicating that they might be better at colonizing a wide range of environments [75]. The vegetation on green roofs is frequently exposed to the combined stress of heat and drought due to the composition of the shallow substrate layers, which will affect their delivered ecosystem services. While numerous studies examined the effect of inoculating green roof substrates with commercially available inoculants to help aid plants in coping with these stressors, there is good evidence that indigenous soil microbial communities are more effective [80,81]. Soil microbes adapted to the edaphic conditions have been found to better improve plant fitness (e.g., [82,83,84]). Considering that many core taxa in this study are stress-tolerant microbes, they plausibly contribute to the resilience and functionality of green roofs in urban environments, which warrants further investigation.

All fungal key taxa in this study are found to be saprobic, plant-pathogenic (endophytic), or both [85]. It is known that endophytic fungi can switch to a pathogenic lifestyle [86] due to, e.g., a host shift, an imbalance in nutrient exchange or microbial interactions [87]. This is interesting, given the fact that the edaphic conditions in the substrate of extensive green roofs can be extreme at moments. Frequent periods of heat and drought in summer, but also floodings during heavy rainfall, most likely induce stress to all members of the biotic component. We suspect that many key fungal taxa respond quickly to the different environmental conditions by altering their lifestyle. For example, Marttinen et al. [88] identified fungal isolates from brown, necrotic parts of mosses on green roofs and inoculated them to spreading earth moss (*Physcomitrella patens*) to investigate pathogenicity. Amongst the most pathogenic fungal strains were species within *Trichoderma*. However, Rumble et al. [40] inoculated green roofs with *Trichoderma*, and while bryophyte coverage was indeed found to be lower in March and July, it increased in January. Switching lifestyles depending on specific environmental conditions would be a plausible explanation for these different observations.

We, however, remain cautious in assigning lifestyles to the fungal key taxa identified in this study. Due to the limited size of the amplicons obtained, analysis of the taxonomic marker to the species level (or lower) is not possible or likely inaccurate. For example, the most abundant fungal ASV in our study is assigned to *Cladosporium* and it contributes to 9.24% of all reads. However, our rarefied dataset includes 11 ASVs belonging to this genus (although their summed contribution was a little more, i.e., 9.87% of all reads). Some species within *Cladosporium* are known endophytes, whether or not parasitic, but many more are saprophytes. Blasting the ASV on GenBank [65] was inconclusive as the amplicon showed 100% identity with multiple species within this genus. Therefore, we recommend that future studies maximize sequence length, by generating full-length amplicons employing high-quality long-read sequencing to ultimately increase taxonomic resolution.

Interested in whether the type of vegetation initially planted on extensive green roofs affects substrate microbial communities, we examined the alpha- and beta-diversity of the prokaryotic and fungal communities. Overall, *Sedum*–moss roofs harbor microbial communities similar to those from *Sedum*–herbs–grasses roofs: none of the investigated alpha-diversity metrics differed between both roof types, and although the composition of prokaryotic and fungal communities is affected by the type of roof, the amount of variation that could be explained by this variable is relatively low: 6.2% and 7.0%, respectively. Although *Sedum*–herbs–grasses roofs have a higher plant richness than *Sedum*–moss roofs, which is also reflected in higher cover percentages of herbs and grasses, a significant number of the present plant species are species that are also found to have colonized *Sedum*–moss roofs (Appendix A). We assume that many of the initially planted species disappeared over time due to the frequent periods of drought and heat arising in the substrate layers. Although we observed thicker substrate layers in *Sedum*–herbs–grasses roofs, the absolute difference is probably too small to be beneficial for the vegetational layer. Furthermore, it remains plausible that, by sampling randomly, we missed some effects of the vegetational layer on the associated substrate microbes. For example, it has been found that species of *Sedum* and species within Asteraceae are associated with different microbial communities in extensive green roofs [50]. Although it went beyond the scope of this study, disentangling the main drivers behind microbial community assemblage in green roof substrates should be prioritized. We recommend that future studies characterize the vegetational layer and the composition of the substrates more comprehensively to examine the deterministic assembly processes in green roof microbial communities. However, our results indicate that stochastic processes [89,90,91] may affect the microbial communities even more. Many key taxa found in this study are commonly detected in aerial samples from urban environments, contributing significantly to the aeromicrobiome. For example, *Cladosporium*, *Alternaria* or *Paraphoma* have been found to be amongst the most common fungal taxa, if not the most dominant ones, in aerial samples as investigated via high-throughput DNA sequencing [92,93,94,95], plausibly due to their high sporulation rates [96]. Also, species within the orders Rhizobiales and Burkholderiales have been identified as some of the most representative bacterial taxa in the air [97,98,99]. Although we did not take aerial samples, we wonder to which extent the aeromicrobiome influences green roof microbial communities, as previously suggested [45]. Many green roofs in this study show distinct prokaryotic and fungal communities (Appendix A), an observation that has also been made in prior research [45]. Considering the frequent periods of drought and heat in the substrates, the microbial community turnover rate due to recolonization can be substantial.

We conclude that green roof substrates harbor dynamic and intriguing microbial communities, warranting further investigation. Future studies should focus on the core microbiota of green roof substrates by employing high-quality long-read sequencing strategies to improve taxonomic resolution. Understanding the core microbiota and their ecological roles will improve our comprehension of the ecosystem services provided by green roofs. To elucidate the primary drivers of microbial community assembly in green roof substrates, we recommend a more comprehensive characterization of the substrate composition and vegetational layer. This includes analyzing plant biomass (roots and shoots) alongside physical and chemical variables to explore correlations between these factors and specific microbial taxa. Additionally, sequencing the aeromicrobiota surrounding green roofs is suggested to assess the influence of stochastic processes on microbial community assembly. A more thorough understanding of the substrate microbiome from extensive green roofs will inform future design recommendations, including plant species selection, substrate composition, and maintenance practices.

## Figures and Tables

**Figure 1 microorganisms-12-01261-f001:**
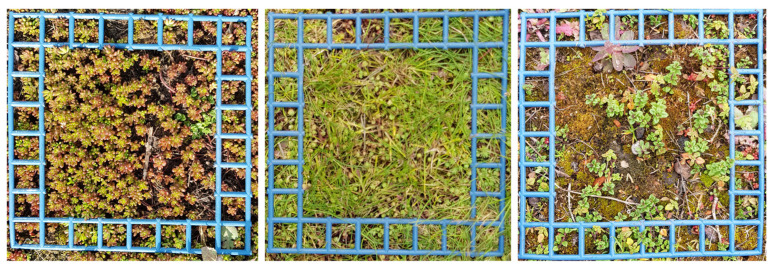
Samples taken during the study. The inner blue squares measure 30 × 30 cm.

**Figure 2 microorganisms-12-01261-f002:**
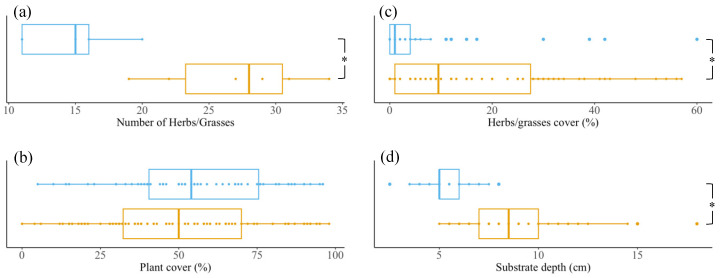
Vegetational characteristics (**a**–**c**) and substrate depth (**d**) of *Sedum*–moss roofs (blue) vs *Sedum*–herbs–grasses roofs (orange). Boxplots span the interquartile range (i.e., the range between the 25th to 75th percentile), and lines within boxes mark the median. Asterisks denote statistically significant differences between both roof types (*p*-value < 0.05; Wilcoxon rank-sum tests).

**Figure 3 microorganisms-12-01261-f003:**
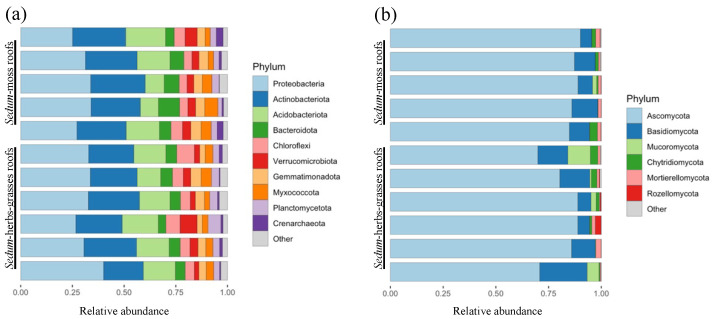
Relative abundances of the 10 major prokaryotic phyla (**a**) and six fungal phyla (**b**) for each investigated extensive green roof (n = 11–15 per roof).

**Figure 4 microorganisms-12-01261-f004:**
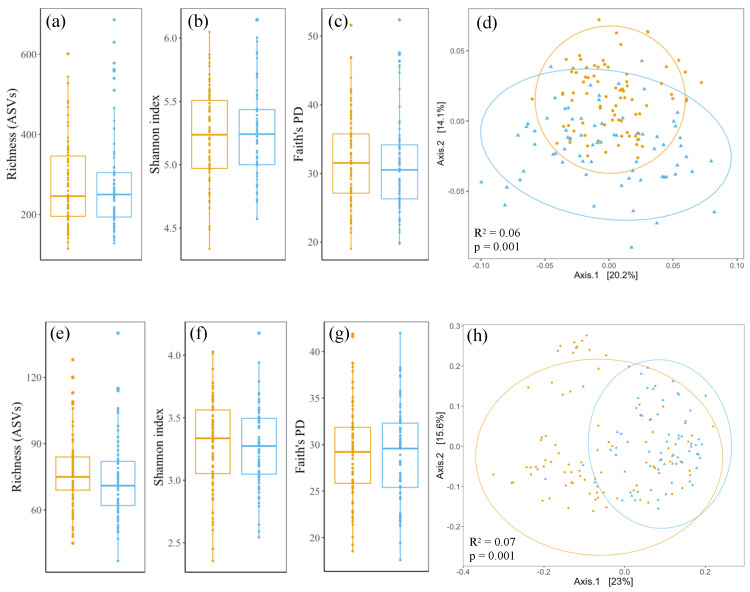
Diversity metrics for prokaryotic (**a**–**d**) and fungal (**e**–**h**) communities in *Sedum*–moss roofs (blue) and *Sedum*–herbs–grasses roofs (orange). Boxplots span the interquartile range (i.e., the range between the 25th to 75th percentile), and lines within boxes denote the median. Circles denote 95% data intervals. No significant differences are found between the alpha-diversity metrics of both roof types (*p*-value > 0.05; Wilcoxon rank-sum test).

**Figure 5 microorganisms-12-01261-f005:**
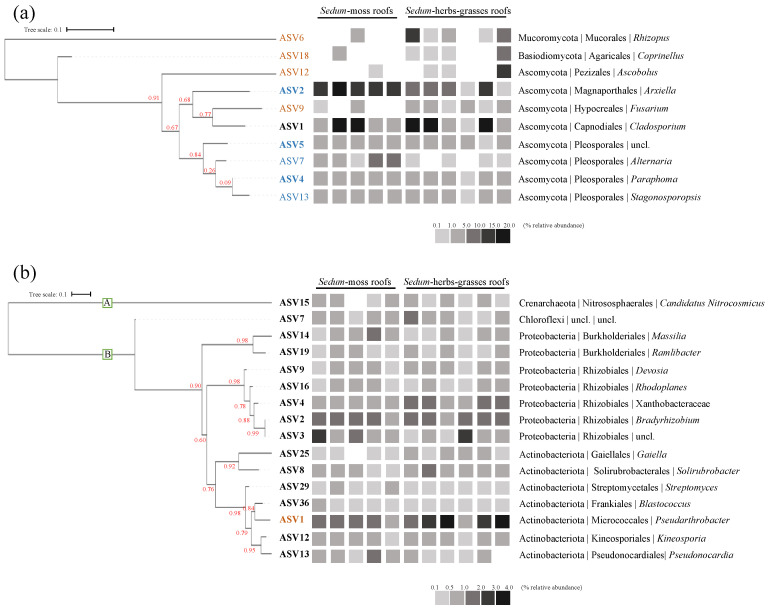
Key fungal (**a**) and prokaryotic (**b**) taxa (ASVs) and their taxonomic classification (phylum |order|genus). Core taxa, having a relative abundance of >0.1% in at least 75% of all samples, are displayed in bold. Microbial ASVs that are more abundantly present, as investigated via LEfSe (*p* < 0.01, LDA log10 scores ≥ 4.0), in *Sedum*–moss roofs are displayed in blue, those in *Sedum*–herbs–grasses roofs in orange. The numbers of the ASVs correspond to their relative abundances in the whole dataset, with 1 being the most abundantly present ASV. The heatmaps show the average relative abundance of each key ASV per roof. Support values for the nodes are displayed in red. A: Archaea, B: Bacteria, uncl: unclassified.

**Table 1 microorganisms-12-01261-t001:** Green roof coordinates and descriptive characteristics. Roof ID, the city in which each roof is located, together with their respective coordinates, their year of construction (Year), the height above ground level at which they are situated (Height), the size of the vegetated area (Area) and the vegetation that was initially planted on them (Type) are summarized.

ID	City	Coordinates	Year	Height (m)	Area (m^2^)	Type
R1	Ghent	51.0239 N 3.7665 E	2014	3.2	25	*Sedum*–herbs–grasses
R2	Ghent	51.0479 N 3.7419 E	2015	3.4	76	*Sedum*–herbs–grasses
R3	Ghent	51.0457 N 3.7509 E	2005	10.5	110	*Sedum*–moss
R4	Ghent	51.0766 N 3.7211 E	2013	8.4	588	*Sedum*–moss
R5	Hasselt	50.9285 N 5.3430 E	2015	7.0	225	*Sedum*–herbs–grasses
R6	Hasselt	50.9338 N 5.3419 E	2012	6.0	81	*Sedum*–moss
R7	Hasselt	50.9263 N 5.3410 E	2004	3.0	175	*Sedum*–herbs–grasses
R8	Antwerp	51.1927 N 4.4163 E	2014	22.3	708	*Sedum*–moss
R9	Antwerp	51.1693 N 4.3941 E	2008	9.0	280	*Sedum*–moss
R10	Antwerp	51.2507 N 4.4190 E	2009	17.4	777	*Sedum*–herbs–grasses
R11	Antwerp	51.2302 N 4.4165 E	2015	9.0	312	*Sedum*–herbs–grasses

## Data Availability

The prokaryotic and fungal metabarcoding samples are uploaded in the Short Read Archive (SRA) of the National Center for Biotechnology Information (NCBI) under the BioProject accession number PRJNA1092393.

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
