# Peer review of "Green Roof Substrate Microbes Compose a Core Community of Stress-Tolerant Taxa"

_microorganisms, 2024, doi:10.3390/microorganisms12071261_

Round 1

Reviewer 1 Report

Comments and Suggestions for Authors

Comments on the Quality of English Language

See attached comments.pdf.

Reviewer 2 Report

Comments and Suggestions for Authors

The authors selected eleven extensive green roofs including planted Sedum species solely or a combination of Sedum to identify the core microbial taxa, and test whether plant diversity affect microbial community composition. I have some specific suggestions in the method and results part. (1) how to keep the sampling method consistent to test the plant diversity influence the microbial community composition. How to calculate the plant diversity? (2) The core and discriminant microbial taxa only identified by its relative abundance, I suggest maybe the network analysis could give more clear key microbial taxa here. (3) The authors should give us which taxa is core or discriminant in the results. 

Specific comments

L84-86, four succeeding seasons only sampled once? This is confusing. 

L106, add the notes of species type including herbs, grasses and moss. In addition, what is the “year” mean? By the way, the height of R9 is a mistake.

L264: More discussion about the Fig 4d and 4h, the reasons of different mechanism in prokaryotic and fungal communities. 

L280-282, should give us the specific taxa of 15 prokaryotic and 14 bacterial core taxa in the text. 

Round 2

Reviewer 1 Report

Comments and Suggestions for Authors

No comment

Author Response

Thank you.

Reviewer 2 Report

Comments and Suggestions for Authors

The authors made some changes according to my comments. However, I still have some suggestions.

1. the authors stated that the goal of this paper was rather to examine the microbial communities from both roof types and investigate whether their overall microbial community compositions vary, if this is correct. Why the authors compare the different roof types and their microbial diversity, core taxa.

2. Indeed, the Lefse analysis could give us the core or discriminant taxa, while the network analysis could give us the correlation between the core taxa.

3. In fig 4d and 4h, the authors presented the figure, and then should give some explanation of this phenomenon, I suggest the authors related to this ref. https://doi.org/10.1128/MMBR.00002-17, which was “Stochastic Community Assembly: Does It Matter in Microbial Ecology?”

Author Response

Dear reviewer,

Thank you for your suggestions.

In attachment you will find our response.

Round 3

Reviewer 2 Report

Comments and Suggestions for Authors

I agree the authors reply.